# Analysis of the Bacterial Microbiota in Wild Populations of Prickly Pear Cochineal, *Dactylopius opuntiae* in Morocco

**DOI:** 10.3390/insects16121184

**Published:** 2025-11-21

**Authors:** Imane Remmal, Youssef El Yamlahi, Naima Bel Mokhtar, Ioannis Galiatsatos, Dimitrios Loukovitis, Eva Dionyssopoulou, Mohammed Reda Britel, Panagiota Stathopoulou, Amal Maurady, George Tsiamis

**Affiliations:** 1Laboratory of Innovative Technologies, National School of Applied Sciences of Tangier, Abdelmalek Essaadi University, Tetouan 93000, Morocco; remmalimane@gmail.com (I.R.); elyamlahi.youssef@gmail.com (Y.E.Y.); mbritel@uae.ac.ma (M.R.B.); amal.maurady.ma@gmail.com (A.M.); 2Faculty of Sciences and Technology of Tangier, Abdelmalek Essaadi University, Tetouan 93000, Morocco; 3Laboratory of Systems Microbiology and Applied Genomics, Department of Sustainable Agriculture, University of Patras, 2 Seferi St, 30100 Agrinio, Greece; naima1503@gmail.com (N.B.M.); jgalia96@gmail.com (I.G.); edionys@upatras.gr (E.D.); panstath@upatras.gr (P.S.); 4Department of Fisheries and Aquaculture, School of Agricultural Sciences, University of Patras, New Buildings, 30200 Mesolongi, Greece; dloukovi@upatras.gr; 5BioDetect P.C., Stadiou St., Platani, 26504 Rio, Greece

**Keywords:** *Dactylopius opuntiae*, biological control, full 16S rRNA gene, symbiosis, *Wolbachia*, *Spiroplasma*, *Candidatus* Dactylopiibacterium, MinION amplicon sequencing

## Abstract

The cochineal *Dactylopius opuntiae* (Cockerell) (Hemiptera: Dactylopiidae) is a significant pest affecting prickly pears in Morocco and globally. This study focused on the bacterial community associated with *Dactylopius opuntiae*. Through targeted PCR screening and high-throughput sequencing of the full-length 16S rRNA gene, we successfully identified low-abundance endosymbionts, specifically *Wolbachia* and *Spiroplasma*, and identified *Candidatus* Dactylopiibacterium as the predominant bacterium. Following the exclusion of this taxon from the dataset during bioinformatic analysis, a more abundant and varied microbiota was observed, highlighting specific patterns related to developmental stage and geographic location. These findings deepen our understanding of the *D. opuntiae* microbiome and provide valuable insights into possible biological control strategies.

## 1. Introduction

*Dactylopius opuntiae* (Cockerell) (Hemiptera: Dactylopiidae), also known as the wild cochineal scale, is a soft-bodied, flat, and oval-shaped parasitic insect that has gained significant attention in the scientific community. It is recognized by immobile females, is apterous, and can reach a length of up to 5 mm [1,2]. *Dactylopius opuntiae* is among the most destructive species in the genus *Dactylopius* [3,4], especially in recently cultivated areas in the Mediterranean region, including France, Spain, Morocco, and Lebanon [5,6,7,8]. This cochineal infestation reduces productivity and renders fruits and cladodes unsellable [9,10]. Scale insects cause a decline in vigor, yellowing of cladodes, fruit drop, and cactus mortality when they become well-established and cover more than 75% of the cladode surface [11,12,13]. With an average of 150–160 eggs laid by females, which rapidly develop into nymphs, infestations are rapid and uncontrollable [9]. In the field, female life cycles range from 40 to 180 days, depending on the season and weather, whereas males typically complete their life cycles in 35 to 52 days [11,14]. During its life cycle, cochineal releases a white waxy covering that envelops its body, reducing the efficiency of phytosanitary treatments [9]. The proliferation of *D. opuntiae* in Mediterranean countries has sparked discussions about the best ways to manage this pest. Currently, chemical and biological approaches are the mainstays of *D. opuntiae* control. Mechanical methods can be used when only a few plants are infested [15,16]. Since its discovery at the end of 2014, *D. opuntiae* has expanded rapidly and caused significant damage in Morocco, prompting local authorities to launch an emergency intervention by removing and burning approximately 400 ha of crops in the Doukkala region [17]. In Morocco, various integrated pest management approaches for cochineal control have been investigated, including host plant resistance, biological control, and biopesticides [18,19,20,21,22,23].

*Dactylopius opuntiae* causes significant agricultural damage worldwide, with infestations in Morocco resulting in the destruction of 400 hectares of crops in the Doukkala region alone [18]. The pest reduces productivity through vigor decline, cladode yellowing, fruit drop, and plant mortality when its coverage exceeds 75% of the cladode surface [11,12,13]. Current control methods include mechanical removal, chemical treatments, and emerging biological control approaches [18,19,20,21,22,23].

Symbiotic microorganisms offer promising avenues for pest management through the disruption of required symbionts or manipulation of pest-relevant traits [24]. Insects harbor diverse microbial communities that influence their nutrition, reproduction, fitness, immunity, and pest status [25,26,27]. Among reproductive symbionts, *Wolbachia* infects 40–75% of arthropod species [28,29] and can manipulate host reproduction through cytoplasmic incompatibility, making it valuable for pest control strategies [30,31]. *Spiroplasma*, found in 5–10% of insect species [29,32,33,34], can cause male-killing or provide protection against stressors [35,36,37,38,39,40,41,42]. Certain *Spiroplasma* species, such as *Spiroplasma poulsonii*, cause male killing in flies, such as *Drosophila* [39,40], *Spiroplasma ixodetis* in butterflies [41], and *Spiroplasma* sp. in ladybird beetles, which alters the sex ratio [42]. Other *Spiroplasma* species, such as *Spiroplasma citri* [41] and *Spiroplasma apis* [43], are known to infect plants and arthropods such as bees [43,44,45,46]. However, some flies infected with *Spiroplasma* may become resistant to other infections [42,47,48,49]. Nonetheless, further research revealed that *Spiroplasma* has various symbiotic relationships [40,47,50,51,52,53].

Research on the microbiota of *D. opuntiae* is scarce and has predominantly concentrated on cultivable microorganisms or populations beyond Morocco. Martínez-Martínez et al. (2024) examined cultivable bacteria associated with carminic acid metabolism in predators of *D. opuntiae* [54].

To the best of our knowledge, this is the first study to examine the bacterial microbiome profile of the full 16S rRNA gene of *Dactylopius opuntiae* located in Morocco using Oxford Nanopore Technology (ONT), Oxford Nanopore Technologies, Oxford, UK, which is one of the most cutting-edge and rapidly developing sequencing technologies [55]. Moreover, this study is the first to report *Wolbachia* infection in *D. opuntiae*, whereas previous findings have only been reported for *Dactylopius coccus*. Recently, several studies have been published on the bacterial symbionts associated with the Mexican carmine cochineal *Dactylopius coccus* (Hemiptera: Coccoidea: Dactylopiidae). This species belongs to the same family (Dactilopiidae) and exhibits morphological characteristics similar to those of *Dactylopius opuntiae* [56]. According to studies conducted by Ramírez-Puebla et al. (2016) and Vera-Ponce De León et al. (2017), the metagenomic approaches employed in their research identified two distinct strains of *Wolbachia* (wDacA and wDacB), a *Spiroplasma*, and a betaproteobacterium that has been designated as *Candidatus* Dactylopiibacterium carminicum [57,58]. However, *Spiroplasma* has been previously reported in this species by Vera-Ponce León et al. (2021), who sequenced and analyzed the genome of a *Spiroplasma* symbiont associated with both *D. opuntiae* and *D. coccus* [59]. Furthermore, researchers have reported the detection of other bacterial species, such as *Massilia*, *Herbaspirillum*, *Acinetobacter*, *Mesorhizobium*, and *Sphingomonas*, which may represent transient gut microbiota acquired from the host plant [60].

Despite the growing interest in the microbiomes of *Dactylopius* species, comprehensive assessments of bacterial diversity across geographical regions remain limited. This study investigated the relationship between the bacterial community profiles of *D. opuntiae* across four distinct regions in Morocco through high-throughput sequencing of the entire 16S rRNA gene, using nanopore technology. Additionally, the presence of reproductive endosymbionts, specifically *Spiroplasma* and *Wolbachia*, was examined, as they may hold potential for microbiome-based biocontrol strategies.

## 2. Materials and Methods

### 2.1. Dactylopius Opuntiae Collection and DNA Isolation

*Dactylopius opuntiae*-infected prickly pear cacti were collected from Agadir, Rabat, Meknes, and Ouazzane, four of Morocco’s principal-producing regions (Appendix A). During the autumn-winter of 2022, *Dactylopius opuntiae* nymphs and adults (males and females) were collected, stored in absolute ethanol, and maintained at −20 °C until use. At certain locations, farmers implemented management practices like burning (Rabat and Meknes) or chemical treatment (Agadir) to manage infestations, whereas other locations remained untreated (Ouazzane). Samples were collected from all sites to offer contextual information regarding the environments; however, the analysis concentrated on geographical variation in the microbiome composition of *D. opuntiae* (Table 1).

Prior to DNA extraction, each sample was surface-sterilized using a 70% *v*/*v* ethanol solution, rinsed with sterile deionized water to eliminate any remaining ethanol, and allowed to dry on a sterile surface. The whole individual fly’s DNA was isolated using a modified CTAB (cetyl trimethyl ammonium bromide) protocol [61]. thereby capturing both gut- and tissue-associated bacteria while minimizing contributions from surface microbes. A Q5000 micro-volume UV–Vis spectrophotometer (Quawell Technology, San Jose, CA, USA) was used to measure the amount and quality of the DNA preparations as well as the concentration of double-stranded DNA. The DNA samples were stored in Eppendorf tubes at −20 °C until further examination. 

### 2.2. Screening and Identification of Bacterial Symbionts

To screen for bacterial symbionts, 120 samples were selected (10 males, 10 females, and 10 nymphs from each region). *Wolbachia* and *Spiroplasma* were detected by PCR using primers specific to the 16S rDNA gene (Appendix A). These two endosymbiotic genera were specifically targeted based on prior investigations in *Dactylopius* species [57,58]. While *Spiroplasma* has been previously reported in *D. opuntiae* [59], *Wolbachia* has not yet been detected in this species. Therefore, our screening aimed both to confirm the presence of *Spiroplasma* and to investigate, for the first time, the potential occurrence of *Wolbachia* in *D. opuntiae*. During DNA extractions, blank and negative controls were included, and the PCRs were performed under the same conditions. However, these samples did not yield any amplicons. Nested PCR amplification was performed in two stages for screening. Using the bacterial universal primers 27F-1492R, the initial amplification was carried out in a 25 µL reaction that included 2 µL of the template DNA solution, 2.5 µL of KAPA Taq buffer 10X, 0.2 µL of dNTPs (25 mM), 0.2 µL of KAPA Taq DNA polymerase (Roche, Basel, Switzerland), 0.6 µL of forward primer (25 M), 0.6 µL of reverse primer (25 M), and 18.9 µL of water. DNA was first denatured for 3 min at 95 °C, followed by twenty cycles of 95 °C for 30 s, 53 °C for 30 s, and 72 °C for 2 min, and a final 5 min extension at 72 °C. The second round of PCR amplification was performed using *Wolbachia*-specific primers (WspecF-WspecR) and *Spiroplasma*-specific primers (Spou R1-Spou F1) in a 25 µL reaction mixture containing 2.5 µL of KAPA Taq buffer 10X, 0.2 µL of dNTPs (25 mM), 0.1 µL of KAPA Taq DNA polymerase, 0.4 µL of the forward primer (25 M), 0.4 µL of the reverse primer (25 M), 1 µL of the first-step reaction as template, and 20.4 µL of sterile deionized water. The PCR amplification was conducted with incubation at 95 °C for 3 min, followed by 20 cycles of 95 °C for 30 s, primer-specific temperature for 30 s, and 72 °C for 1 min, with a final 5 min extension at 72 °C. The sizes of the amplified fragments were assessed by electrophoresis of the PCR products on a 1.5% agarose gel. Appendix A summarizes the product sizes, annealing temperatures, and primer sequences used in this study. Following purification with polyethylene glycol (20% PEG, 2.5 M NaCl) [62], PCR-positive products were resuspended in 15 µL water. Sanger sequencing was performed on the purified products using the BigDye Terminator v3.1 Cycle Sequencing Kit according to the manufacturer’s instructions (Applied Biosystems, Waltham, MA, USA). Reaction products were purified using an ethanol/EDTA process following the manufacturer’s recommendations (Applied Biosystems, Waltham, MA, USA) and sequenced on an ABI PRISM 3500 Genetic Analyzer (Applied Biosystems, Waltham, MA, USA).

### 2.3. Amplification of the 16S rRNA Gene, Library Preparation, and MinION Sequencing

For MinION amplicon sequencing, 73 samples were selected (Table 1). Using 27F-1429R primers, the full 16S rRNA gene was amplified (Appendix A). PCR amplification was conducted in a 25 µL reaction comprising 2.5 µL of KAPA Taq buffer 10**X**, 0.2 µL of dNTPs (25 mM), 0.2 µL of KAPA Taq DNA polymerase (Roche, Basel, Switzerland), 0.6 µL of forward primer (25 M), 0.6 µL of reverse primer (25 M), 1 µL of the template DNA solution, and 19.9 µL of water. A 5 min incubation time at 95 °C was used for DNA denaturation, followed by 35 cycles of 95 °C for 30 s, 54 °C for 30 s, and 72 °C for 2 min, with a final 5 min extension at 72 °C. For the MinION sequencing, the library was prepared using the SQK-NBD114.96 kit from Oxford Nanopore Technologies (Oxford, UK) [63]. Following the manufacturer’s instructions, PCR-purified products were initially diluted to a concentration of 250 fmol in a final volume of 11.5 µL and sequenced. End-preparation was completed by adding a reaction buffer-enzyme mix and 1 µL of a diluted DNA Control Sample (DCS). A thermal cycler was used to incubate the reaction mixture at 20 °C for 5 min and 65 °C for 5 min. Next, 0.75 µL of End-prep-DNA was mixed with 3 µL of sterile distilled water (SDW), 1.25 µL of Native Barcode (NB01-96), and 5 µL of Blunt/TA Ligase Master Mix. Regarding the barcoding, a purification process was carried out to eliminate any unincorporated chemicals. The DNA library (5 µL) was combined with 1.5 µL of sequencing buffer and 10 µL of loading beads before loading onto a FLO-MIN114 flow cell. Sequencing was performed using a MinION MK1B device, and data acquisition was controlled using MINKNOW software version 23.11.5 (Oxford Nanopore Technologies).

### 2.4. Bioinformatics Analysis

The basecalling for the DNA sequence analysis was carried out with Dorado (version 0.8.2) [64], a highly effective tool created by Oxford Nanopore data. The samples were then demultiplexed using Poreshop (https://github.com/rrwick/Porechop, accessed on 15 November 2025) to identify and remove the barcode sequences from the reads [65]. Raw reads were filtered using NanoFilt (version 2.8.0) according to their length (from 1200 to 1600 bp) and quality (Qscore > 9) [66]. De novo clustering, consensus building, and polishing were performed using the NanoClust (https://github.com/genomicsITER/NanoCLUST, accessed on 15 November 2025) pipeline [67]. Taxonomy was performed with Qiime2 with the BLAST + (version 2.14.1) algorithm against the SILVA 138.2 release database [68,69]. Diversity within samples was determined using alpha diversity measures: richness, evenness, Shannon, and Simpson indices. Nonparametric Kruskal–Wallis and Wilcoxon rank-sum tests were used to assess statistical differences in bacterial abundance between populations [70]. To display the similarities between bacterial communities in different areas, beta diversity analysis was conducted based on the Bray–Curtis distance. This was visualized using Canonical Analysis of Principal Coordinates (CAP) [71]. To test for significant differences between the studied categories, a permutational multivariate analysis of variance (PERMANOVA) was performed. Statistical significance was considered as a *p*-value of <0.05. Additionally, Core microbiome analyses were performed using MetaXplore (version 1.0) [72] to identify the major bacterial taxa consistently present across samples. OTUs were considered part of the core if they were detected in ≥50% of the samples within each region (prevalence threshold) and had a relative abundance ≥0.1% (detection threshold). In contrast, MetaXplore [72] was also used for visualization and downstream analyses, including alpha diversity, beta diversity, and relative abundance. All data supporting the findings of this study are accessible in the NCBI under BioProject PRJNA1293817.

### 2.5. Phylogenetic Analysis

Phylogenetic analysis was performed using partial 16S rRNA gene sequences derived from specimens infected with *Wolbachia* and *Spiroplasma*. Multiple alignments were performed using MUSCLE, as implemented in MEGA 11 software, using standard parameters [73,74]. The sequence length was adjusted by manual editing and trimming of the alignment. The maximum likelihood statistical approach was used to reconstruct the phylogenetic tree using MEGA 11 software. Nucleotide evolution was estimated using the (GTR + G + I) substitution model [75,76]. All 16S rRNA gene sequences generated in this study were uploaded to the NCBI GenBank database with accession numbers for *Wolbachia* sequences from PV089130-PV089147 and *Spiroplasma* sequences from PV089148-PV089151.

## 3. Results

### 3.1. Infection Status of Reproductive Symbionts in Natural Populations of D. opuntiae

#### 3.1.1. Infection Prevalence of Endosymbiotic Bacteria

PCR screening was performed to investigate the presence of endosymbionts, *Wolbachia* and *Spiroplasma*, across four natural *D. opuntiae* populations. A total of 120 samples were analyzed (Appendix A). The screening results indicated that *D. opuntiae* samples were infected with *Wolbachia,* with a prevalence of 16.6%. Interestingly, the percentage of *Wolbachia* infection in natural populations was not evenly distributed across different locations. In total, 16 samples were infected, six nymphs, four females, and six males out of 30 samples examined from Ouazzane (53.5%), one female out of 30 from Meknes (3.33%), and one female out of 30 samples examined from Agadir (3.33%), while no infection was found in the Rabat region (Appendix A). The *D. opuntiae* populations examined were also infected with *Spiroplasma*, with a prevalence of 3.33% across all regions, equivalent to four out of 120 samples. Three out of 30 males from Agadir (10%) and one male out of 30 samples from Ouazzane (3.33%). In contrast, samples from the Meknes and Rabat populations were not infected with *Spiroplasma* (Appendix A).

#### 3.1.2. Phylogenetic Analysis of *Wolbachia* and *Spiroplasma* Sequences in *D. opuntiae* Populations

*Wolbachia* phylogenetic analysis was performed on 18 *Wolbachia*-infected samples based on partial 16S rRNA gene sequences, using a total of 311 bp of high-quality sequences retained after manual trimming of low-quality ends. The results showed that the *Wolbachia* sequences detected in *D. opuntiae* populations belonged to supergroup B, exhibiting a high sequence similarity of pairwise distances (98%) with *Wolbachia* sequences isolated from *Sitophilus oryzae* species (Figure 1).

Phylogenetic analysis of *Spiroplasma* was performed on the four *Spiroplasma*-infected samples based on partial 16S rRNA gene sequences, using a total of 349 bp of high-quality sequence retained after manual trimming of low-quality ends. According to the results, the *Spiroplasma* sequences detected in *D. opuntiae* populations belonged to the *Spiroplasma* poulsonii–citri complex, exhibiting a high pairwise distance (between 98 and 99%) (Figure 2).

### 3.2. 16S rRNA Amplicon Sequencing

The bacterial community composition and diversity of 73 wild *D. opuntiae* samples from the Agadir, Rabat, Meknes, and Ouazzane regions were investigated using full-length 16S rRNA gene. After sequencing and quality filtering, 1,103,422 qualified reads were generated, with an average of 13,456 reads/sample. Sixty-two clusters were classified into five phyla, with Pseudomonadota being the most dominant (98.5%), followed by Bacillota, Bacteroidota, Cyanobacteriota, and Actinomycetota. Seven classes were identified: Gammaproteobacteria was the dominant class, comprising 98.2% of the bacterial community, followed by Alphaproteobacteria, Bacilli, Chitinophagia, Actinobacteria, Negativicutes, and Cyanobacteriia. At the genus level, seven genera were identified across all samples, with *Uliginosibacterium* (*Candidatus* Dactylopiibacterium) as the most abundant genus, representing 97.7% of the bacterial community, whereas the rest of the genera represented less than 1% across all regions (Appendix A). 

#### 3.2.1. Bacterial Diversity and Composition Among *D. opuntiae* Natural Populations

The bacterial community of *D. opuntiae* varied across different regions, highlighting the influence of geography on microbial structure. Alpha diversity analysis revealed significant regional variations. Meknes showed the highest Shannon and Simpson diversity indices, which were significantly higher than those of Agadir (Tukey HSD *p* < 0.05). Agadir exhibited slightly higher richness than the other regions, although the differences were not significant (Appendix A). Beta diversity analysis based on Bray–Curtis distances and PERMANOVA revealed significant compositional differences among regions (global *p* < 0.008) (Figure 3A). PERMANOVA analyses revealed significant differences among regions: Agadir and Meknes (*p* = 0.001), Meknes and Rabat (*p* = 0.013), and Agadir and Rabat (*p* = 0.005), indicating spatial variation in microbial communities (Figure 3B). However, Ouazzane showed no significant difference from the other regions (Figure 3B). After excluding the dominant *Candidatus* Dactylopiibacterium, no significant difference was observed between the regions. This indicates that the observed regional variation was mainly driven by differences in the relative abundance of this dominant symbiont (Appendix A). Bacterial communities from the chemically treated region (Agadir) exhibited similarities to those from the untreated region (Ouazzane), indicating no effect of chemical treatment on microbiome composition. Bacterial communities in burned regions (Rabat and Meknes) differed from those in the chemically treated region (Agadir), indicating that management practices may affect microbial composition. Differences between the two burned regions suggest that geographical factors significantly influence the composition of *D. opuntiae* bacterial communities.

Pseudomonadota was the most prevalent phylum identified across all regions (Agadir, Rabat, Meknes, and Ouazzane) with relative abundances ranging from 99.642 ± 0.11% to 96.172 ± 3.64%, respectively. Bacillota was the second abundant phylum in Meknes, Rabat, and Agadir (from 0.543 ± 0.32% to 0.238 ± 0.08%, respectively) In Ouazzane the second phylum was Bacteroidota (3.77 ± 3.64%), followed by Bacteroidota which was found in lower abundance in Meknes, Agadir, and Rabat (from 0.425 ± 0.12% to 0.085 ± 0.04%, respectively). Other phyla, such as Cyanobacteriota and Actinomycetota, were only found in samples from Rabat, although their abundance was still relatively low (0.464 ± 0.45% and 0.024 ± 0.02%, respectively) (Appendix A). At the class level (Appendix A), Gammaproteobacteria was the most dominant class in Agadir, Meknes, Rabat, and Ouazzane (from 99.37 ± 0.11% to 95.986 ± 3.63%). While the remaining classes were distributed with varying prevalences across all regions, the case of Meknes, the second class was Bacilli, followed by Chitinophagia, Alphaproteobacteria, and Negativicutes (from 0.526 ± 0.31% to 0.017 ± 0.01%, respectively). In contrast, the second class in Ouazzane was Chitinophagia, followed by Alphaproteobacteria and Bacilli (from 3.77 ± 3.64% to 0.058 ± 0.04%, respectively). In Agadir, the second class was Alphaproteobacteria, followed by Bacilli and Chitinophagia (from 0.272 ± 0.1% to 0.12 ± 0.07%, respectively). Conversely, in Rabat, the second most abundant class was Cyanobacteria, followed by Bacilli, Alphaproteobacteria, Chitinophagia, Actinobacteria, and Negativicutes (from 0.464 ± 0.45% to 0.012 ± 0.01%, respectively) (Appendix A). At the genus and species levels, *Uliginosibacterium* dominated across all locations, represented by *Candidatus* Dactylopiibacterium (from 98.961 ± 0.21% to 95.544 ± 3.61%), followed by a low abundance of the genus *Flavisolibacter*, represented by *Flavisolibacter longurius* in the Ouazzane and Meknes regions (3.77 ± 3.64% and 0.425 ± 0.12%, respectively). In Rabat, *Macrochaete*, represented by *Macrochaete psychrophila,* was the second genus, whereas in Agadir, the second genus was *Pseudomonas* , represented by *Pseudomonas* sp. (0.464 ± 0.45% and 0.238 ± 0.08%, respectively), while the rest of the genera were found at low prevalence rates across all regions (<0.1%) (Appendix A).

#### 3.2.2. Bacterial Diversity and Composition Among *D. opuntiae* Gender and Developmental Stage, Excluding *Candidatus* Dactylopiibacterium

The dominance of *Candidatus* Dactylopiibacterium can obscure the presence of other microbial taxa and potentially confound assessments of overall diversity. To analyze the heterogeneity in bacterial structure between *D. opuntiae* gender and developmental stage, clusters corresponding to the dominant *Candidatus* Dactylopiibacterium species were removed. A total of 24 clusters were excluded from the investigation, reducing the number of clusters from 62 to 38. 

##### Bacterial Diversity

Alpha diversity indicators such as ACE richness, Shannon diversity, Simpson’s index, and Pielou’s evenness differed between the seven groups. Interestingly, females from Agadir (F_M) had the lowest richness and diversity across all indices, indicating a limited number of bacterial species. In contrast, males from Meknes (M_M) and nymphs from Rabat (N_R) were more diverse. The data showed that developmental stage and geographic origin shaped the microbial community structure and diversity (Appendix A). 

Beta diversity analysis based on Bray–Curtis dissimilarity demonstrated variations in microbial communities across different developmental stages (Figure 4 and Appendix A). PERMANOVA indicated notable variation in Meknes between females, males, and nymphs (*p* < 0.05), as well as between females and nymphs in Agadir (*p* < 0.05). In contrast, Rabat and Ouazzane exhibited no significant differences across the developmental stages and genders (*p* > 0.05) and were thus combined (Appendix A). In light of these findings, the samples could be grouped into seven distinct categories: Meknes: F_M, M_M, N_M; Agadir: F_A, N_A; Rabat and Ouazzane (Figure 4).

##### Bacterial Composition

While the beta diversity analyses supported grouping samples into seven categories, the relative abundance analysis was conducted according to the developmental stage across each region individually. This approach was adopted to uncover stage-specific differences in bacterial composition that might be obscured by grouped data.

At the phylum level, Pseudomonadota emerged as the predominant bacterial phylum, showing significant dominance in the nymph and male samples. The community comprised more than 90% of nymphs from Agadir (N_A: 93.41 ± 0.7%) and Meknes (N_M: 94.24 ± 0.32%), and was notably prevalent in male samples, attaining 78.40 ± 6.68% in M_O and 77.78 ± 6.17% in M_R. In contrast, the samples from females exhibited greater diversity at the phylum level. The second most prevalent phylum was Bacteroidota, which was particularly abundant in female samples, where it constituted 64.88 ± 9.48% in F_M, 33.23 ± 12.96% in F_O, and 28.08 ± 13.11% in F_R. Bacteroidota was detected at significantly reduced levels in male and nymph samples, especially in N_M at 2.15 ± 0.14% and N_A at 0.71 ± 0.37%. Other phyla, such as Bacillota, were significantly present in F_A at 37.41 ± 9.05% and M_M at 23.94 ± 8.06%, whereas Cyanobacteriota was infrequently observed, reaching a peak in N_R at 8.80 ± 8.77%. The abundance of Actinomycetota was generally low, peaking at 1.52 ± 0.59% in N_A (Figure 5A).

At the class level, Gammaproteobacteria was the predominant bacterial class at various developmental stages, especially in nymphs and males. The highest levels were recorded in N_M (69.98 ± 2.09%), M_O (57.77 ± 4.78%), N_R (51.9 ± 7.11%), and N_O (51.249 ± 8.08%), with dominance observed in females from Rabat (F_R: 43.85 ± 11.26%) and Ouazzane (F_O: 40.01 ± 10.68%). In contrast, females from Meknes (F_M) presented Chitinophagia as the predominant class, comprising 64.88 ± 9.48% of the microbial community. This class was the second most abundant in F_O (33.23 ± 12.96%), in F_R (28.08 ± 13.11%), and in F_A (21.38 ± 8.35%), whereas it had a low prevalence in nymphs and males. In the majority of nymphs and males, Alphaproteobacteria was the second most prevalent group, exhibiting significant values of N_A (34.21 ± 3.12%), M_R (30.11 ± 4.3%), and N_M (24.27 ± 2.38%). In females, Bacilli made a significant contribution, following Chitinophagia and Gammaproteobacteria, particularly in F_A at 34.29 ± 8.33%. Classes such as Negativicutes, Actinobacteria, and Cyanobacteria exhibited low or sporadic abundance, with a significant peak of Cyanobacteria observed in N_R at 8.80 ± 8.77% (Figure 5B).

At the genus level, *Flavisolibacter* (*Flavisolibacter longurius*) was the most abundant genus in all female groups, except for F_A, reaching 64.88 ± 9.48% in F_M, 33.23 ± 12.96% in F_O, and 28.08 ± 13.11% in F_R. In contrast, *Brevibacillus* was the most dominant genus in F_A (27.41 ± 6.85%), making it the top genus in this group. The second most abundant genus across female samples varied: *Pseudomonas* presented by three species, the case for *Pseudomonas* sp., *Pseudomonas mosselii*, and *Pseudomonas putida* was prominent in F_R (23.44 ± 7.06%), F_O (21.13 ± 5.14%), and F_M (13.092 ± 5.24%), respectively. While *Massilia* (7.13 ± 2.39%) and *Paenibacillus* (*Paenibacillus castaneae*) (6.02 ± 1.88%) were notable in F_A. In male and nymph samples, *Pseudomonas* consistently dominated, especially in N_M (43.46 ± 1.81%), N_A (40.72 ± 3.13%), and M_O (35.95 ± 5.14%), followed by *Phyllobacterium* represented by two species, the case for *Phyllobacterium* sp. and *Phyllobacterium myrsinacearum*, *Stenotrophomonas* (*tenotrophomonas maltophilia*), or *Brucella* (*Ochrobactrum anthropi*). *Phyllobacterium* reached 17.73 ± 3.7% in M_R and 22.09 ± 2.99% in N_A, whereas *Stenotrophomonas* peaked at 10.26 ± 1.45% in N_M. Other genera, such as *Acinetobacter* (*Acinetobacter junii*), *Achromobacter* (*Achromobacter denitrificans*), and *Enterobacter* (*Enterobacter cloacae*), were present in lower abundance but contributed to group-specific differences (Figure 5C).

##### Core Microbiome

Regarding the results based on developmental stage and region parameters, the core bacterial community was composed of 24 clusters distributed across the regions/developmental stages (Appendix A). However, except for *Candidatus* Dactylopiibacterium, no specific taxon was shared among all the groups. The most significant intersection occurred between M_M and N_A, sharing 8 genera: *Acinetobacter* [C0], *Aromatoleum* [C28], *Uliginosibacterium* [C37], *Phyllobacterium* [C39], *Brucella* [C4], *Stenotrophomonas* [C48], *Phyllobacterium* (variant) [C54], and *Paenibacillus* [C63]. M_M alone harbored five unique genera: *Sphingomonas* [C11], *Massilia* [C42], *Brevibacillus* [C67], and two distinct clusters of *Flavisolibacter* [C79, C83]. F_A contributed to one unique genus, *Brevibacillus* [C40], whereas F_M had a single core genus, *Flavisolibacter* [C91]. Three genera of *Pseudomonas* were shared by F_A, M_M, and N_A in the case of [C22, C75, C31]. One genus, *Flavisolibacter* [C91], was shared among F_A, F_M, and M_M (Appendix A).

## 4. Discussion

This study provides novel insights into the bacterial community structure associated with *D. opuntiae*, emphasizing the potential relationships between host-associated microbiota, symbiotic associations, and ecological adaptation across geographic regions and developmental stages. These findings enhance our understanding of the composition and variability of *D. opuntiae*’s microbiome and highlight its possible relevance in host fitness and the development of environmentally sustainable control strategies.

### 4.1. Infection Prevalence

While Sanger sequencing identified *Wolbachia* in several individuals of both sexes, *Spiroplasma* was found solely in a specific group of males. This sex-specific pattern may result from random variation due to low infection prevalence or could indicate a biological association between *Spiroplasma* infection and male hosts. Further investigation would be necessary to determine whether this sex-specific distribution reflects a true biological pattern or is due to limited sample detection. However, neither of these endosymbionts was identified using MinION amplicon sequencing. This disparity likely arises from their irregular distribution and low abundance, which may render species undetectable using amplicon-based community profiling. Recent studies by Marshall et al. (2024) [77] and Nolan et al. (2025) [78] demonstrated that nanopore sequencing surpasses Sanger sequencing in elucidating community diversity, especially in identifying co-occurring taxa within mixed samples. This suggests that dominant and prevalent taxa establish a stable microbial community, whereas rare endosymbionts contribute to individual variation. The data indicate that high-throughput sequencing can identify dominant species; however, without adequate sequencing depth or targeted enrichment, it may fail to detect rare or unevenly distributed symbionts [77,78]. In this study, Sanger sequencing was conducted using genus-specific primers for *Wolbachia* and *Spiroplasma* through nested PCR amplification, facilitating the identification of these symbionts, even at low abundance. This highlights how a targeted versus untargeted methodological design can strongly influence the sensitivity of detection, especially for rare or inconsistently distributed taxa. 

To the best of our knowledge, this is the first report of *Wolbachia* in *D. opuntiae*. This finding expands the known host range of symbionts within this genus. *Wolbachia* infection in scale insects of the genus *Dactylopius* was first documented in 2007 [79]. Later, Ramírez-Puebla et al. (2016) provided a detailed characterization and identified two different strains, wDacA (*Candidatus* Wolbachia bourtzisii) and wDacB (*Candidatus* Wolbachia pipientis), in *D. coccus* populations using PCR and metagenomic sequencing [57]. The detection of *Wolbachia* in *D. opuntiae* samples is therefore aligned with earlier work on *D. coccus*, yet it represents a novel association for this pest species. Nonetheless, a variety of agricultural pests are recognized as carrying one or more strains of *Wolbachia*, and the estimated frequency of infection in arthropod species varies between 40 and 75% [28,29], including aphids [61,80,81], multiple species of the Drosophilidae family [82,83,84,85], and fruit flies from the Tephritidae family [86,87,88,89,90,91]. 

Phylogenetic analyses of *Wolbachia* sequences identified in this study demonstrated the highest homology to strains from supergroup B. These findings are in contrast to those of Ramírez-Puebla et al. (2016), who reported that metagenome analysis recovered the genome sequences of *Candidatus* Wolbachia bourtzisii wDacA, identified in *Dactylopius coccus*, which belongs to supergroup A [57]. This divergence likely reflects host and geographical variation, as the distribution of *Wolbachia* supergroups differs across regions [92]. *Wolbachia* density is also known to influence cytoplasmic incompatibility (CI), with low prevalence reducing CI expression and higher densities enhancing it [93,94,95]. Environmental factors such as temperature can further modulate this effect [93]. Given the low infection rate observed, it is unlikely that *Wolbachia* induces CI or alters the sex ratios of *D. opuntiae*. The presence of the symbiont in both sexes suggests stable maintenance, potentially supported by horizontal transmission, a phenomenon frequently observed in insects and possibly mediated by parasitoids or host plants [96,97,98,99]. Nonetheless, the ability of *D. opuntiae* to host *Wolbachia* provides a basis for potential transinfection strategies, which have already been demonstrated in *Drosophila* [100,101], *Ceratitis capitata* [90,102,103,104], *Aedes aegypti* [105,106,107], *Bactrocera oleae* [108], and *Culex quinquefasciatus* [109].

The identification of *Spiroplasma* in Moroccan *D. opuntiae* samples aligns with previous research documenting the presence of this symbiont in *D. coccus*. Ramírez-Puebla et al. (2016) and Vera-Ponce De León et al. (2017) analyzed *Spiroplasma* in *Dactylopius* species using metagenomic techniques [57,58]. More recently, high-quality *Spiroplasma* genomes have been obtained for both *D. opuntiae* and *D. coccus* through genomic analyses (Vera-Ponce León et al., 2021) [59]. In our study, *Spiroplasma* was found exclusively in males, and phylogenetic analysis placed the sequences within the poulsonii–citri clade. This contrasts with Vera-Ponce León et al. (2021), who identified *S. ixodetis* in *D. opuntiae* and *D. coccus* [59]. The identification of the poulsonii clade is notable, as members are known for reproductive manipulation, including male-killing in *Drosophila* [110,111,112], while the identification of citri-clade strain is not related to the gender, but represented as a vector insects that activate hexamerin-mediated immunity [113], and in insects like *Drosophila*, *S. citri* proliferates could causes death, overriding immune defenses [114]. The absence of documented phenotypes in *Dactylopius*, coupled with the unique occurrence of *Spiroplasma* in males, prompts an inquiry into the potential host-symbiont dynamics and ecological significance of this symbiont.

### 4.2. Dynamics of the Bacterial Communities Associated with D. opuntiae Across Geographical Locations

To examine the influence of geography on bacterial populations, 16S rRNA amplicon sequencing was performed on natural *D. opuntiae* samples. Distinct community clusters were identified across the four populations, marking the initial thorough evaluation of bacterial diversity in *D. opuntiae* across several areas and at different developmental stages. Excluding *Candidatus* Dactylopiibacterium from the bioinformatic analyses facilitated the characterization of the remaining low-abundance taxa, which, despite constituting only approximately 2.3% of the community, may impact host fitness and adaptation [115,116]. Comparisons that include and exclude this dominant symbiont further highlighted its influence on overall diversity patterns. Bacterial assemblages exhibited variability in richness and evenness, with certain locations displaying balanced communities, whereas others were dominated by a limited number of taxa. The disparities were evident in both alpha and beta diversity indices, highlighting the significant influence of geography on the microbial composition. Comparable patterns have been observed in other insects; in the case of aphids belonging to the subfamily Hormaphidinae, the dominance of *Buchnera* led to reduced alpha diversity [117]. In parasitoids such as *Dipterophagus daci*, *Wolbachia* dominance reduced evenness, yet beta diversity indicated population-level variation [118]. 

Regarding the bacterial composition of *D. opuntiae,* our results indicated that *Candidatus* Dactylopiibacterium (Betaproteobacterium, Rhodocyclaceae) was predominantly present in *D. opuntiae*, exhibiting an average relative abundance of approximately 97% across all regions, whereas other genera accounted for less than 3%. This consistent prevalence suggests that it is a significant symbiont of *D. opuntiae.* Prior research has indicated its occurrence in various *Dactylopius* species, such as *D. coccus* and *D. opuntiae*, where it is linked to nitrogen recycling and amino acid biosynthesis [58,60]. Detection has occurred in the ovaries of both species [60], and genomes have been sequenced from both sexes of *D. coccus* and wild *D. opuntiae* [58]. It is phylogenetically classified within Betaproteobacteria, order Rhodocyclales, and is closely related to *Azoarcus*, a nitrogen-fixing plant endophyte [119]. Metatranscriptomic analyses have indicated significant transcriptional activity in the hemolymph, along with additional activity in the gut and ovaries, implying a role in nutrient supply and digestion of cactus polysaccharides [120]. These functional capabilities highlight *Candidatus* Dactylopiibacterium as a crucial vertically transmitted symbiont essential for host nutrition, development, and ecological adaptation.

Our findings suggest that geographical variation is the main factor influencing the bacterial communities of *D. opuntiae*, whereas management practices could produce local effects. Bacterial communities in the chemically treated region (Agadir) exhibited similarities to those in the untreated region (Ouazzane), indicating minimal impact of chemical treatment on microbiome composition. Burned regions (Rabat and Meknes) exhibited distinct bacterial communities relative to the chemically treated site, suggesting that local management practices may influence microbial composition. The observed differences between the two burned regions underscore the significant influence of geographical and environmental factors on the composition of microbial communities. The findings align with prior research indicating that insect microbiomes are significantly affected by environmental and ecological factors, including habitat and local conditions [121,122]. In summary, although treatment can influence local microbial profiles, broader spatial variation seems to be the primary factor shaping the bacterial community structure in *D. opuntiae*.

### 4.3. Gender-Based Differences in Bacterial Composition of D. opuntiae

The microbial community of *D. opuntiae* varied based on developmental stage and gender. Rabat and Ouazzane displayed stable profiles across all stages, whereas Meknes and Agadir showed notable differences among females, males, and nymphs. Similar developmental stage-dependent changes have been observed in other insects, such as *Bactrocera dorsalis*, where pupae and adults harbored distinct communities compared to larvae [123]. These results further support this observation by showing that in *D. opuntiae*, adult females exhibit significantly different microbial communities compared to males and nymphs, suggesting a sex-specific pattern that may be shaped by host biology. 

At the genus level, females from Meknes, Ouazzane, and Rabat exhibited a predominance of *Flavisolibacter*, whereas those from Agadir were characterized by *Brevibacillus* as the most prevalent genus, indicating significant geographical differences. Males and nymphs exhibited a consistent dominance of *Pseudomonas* across the regions, indicating a stable association of this genus regardless of the region and developmental stages. The ecological roles of these taxa likely contribute to their prevalence. For example, *Flavisolibacter*, a genus associated with soil and plants, exhibits chitin-degrading capabilities and plays a role in organic matter turnover [124,125,126,127,128]. *Brevibacillus* serves both entomopathogenic and symbiotic functions in insects [129,130,131], and *Pseudomonas* is recognized for its wide range of habitats and hosts, as well as its notable metabolic diversity [132,133]. This genus has evolved to engage in both beneficial and detrimental interactions, primarily with plants [134,135], insects [136,137], and humans [138]. Less abundant genera, such as *Phyllobacterium* and *Stenotrophomonas*, likely reflect transient associations acquired from host plants or the surrounding environment, rather than stable symbiotic relationships within the insect. The observed patterns indicate that the *D. opuntiae* microbiome is influenced by intrinsic factors such as sex and developmental stage, as well as extrinsic factors including geographic region and host plants, demonstrating a flexible and adaptive microbial community.

Our findings also identified *Massilia* (Oxalobacteraceae), represented by the *Massilia oculi* species, a genus noted for its extensive ecological distribution. It has been isolated from various environments, including air, aerosols, dust, water, soil, the phyllosphere, the rhizosphere, and the roots of multiple plant species [139,140,141], as well as from insects [60]. *Massilia* can be selectively enriched through root exudates, which affect plant metabolism by improving nitrogen uptake and auxin-related pathways, consequently promoting plant growth and enhancing seed oil content [142]. *Acinetobacter*, represented by *Acinetobacter junii*, is typically recognized as a common free-living saprophyte that exhibits significant metabolic versatility and the ability to adapt to various human-associated and natural environments [143,144,145]. Reports indicate its presence in agricultural soils, seawater, plants, and insects [146,147,148], where it is functionally associated with the digestion of plant polymers [149], detoxification of plant compounds, and immune defense mechanisms [150]. *Sphingomonas*, represented by *Sphingomonas echinoides* species, has been reported to contribute to plant protection and the enhancement of plant growth [151,152], as well as its involvement in the bioremediation of environmental contaminants [153,154] and stress tolerance [155], isolated from a range of sources, including marine water [156], soil [157], and insects [158,159]. Interestingly, these three genera (*Massilia*, *Acinetobacter*, and *Sphingomonas*) were also identified by Ramírez-Puebla et al. (2010) [60] in various *Dactylopius* species, including *D. opuntiae*. The observed overlap indicates a possible horizontal transmission of these bacteria from Cactaceae plant sap to *Dactylopius* spp. during feeding [60]. 

Core microbiome analysis did not reveal any common genera across all developmental stages and location groups, indicating the lack of a universal core in *D. opuntiae*. Overlaps were specific to groups, with males from Meknes (M_M) and nymphs from Agadir (N_A) exhibiting the greatest number of shared genera. In contrast, genera such as *Flavisolibacter*, *Brevibacillus*, and *Sphingomonas* were associated with specific stages or regions. The consistent occurrence of *Pseudomonas* and *Flavisolibacter* across various groups indicates that these taxa are likely to fulfill important functional roles within the host’s microbiome. The integration of taxonomic and diversity analyses indicated a sex-specific organization of the microbiome, implying that females possess a more specialized and potentially functionally distinct bacterial community. Gao et al. (2022) and Wang et al. (2023) observed a comparable sex-specific core microbiome pattern in the wolf spider *Pardosa astrigera*, with females containing only three unique OTUs, whereas males exhibited over 110 male-specific OTUs, along with 155 shared between the sexes [160,161]. Similarly, in the invasive mealybug *Phenacoccus solenopsis*, Wang et al. (2023) observed notably different core microbiomes between females and males [161].

Overall, our results demonstrate the intricate bacterial community structure of *D. opuntiae*, which is influenced by host sex and geographic origin. Targeted Sanger sequencing uncovered low-abundance symbionts (*Wolbachia* and *Spiroplasma*) that remained undetected by MinION amplicon sequencing, highlighting the significance of employing complementary methods. The elimination of the predominant *Candidatus* Dactylopiibacterium revealed a more diverse microbiota, with *Flavisolibacter* identified as a significant taxon that varied by region and consistently dominated the bacterial community of females, suggesting pronounced sex- and geography-related structuring. Females demonstrated a more limited core microbiome, indicating a unique microbial pattern that was linked to sex. The microbial variation in *D. opuntiae* is influenced by developmental stage, sex, geographic origin, environmental exposure, host plant interactions. Additionally, differences in microbial profiles may reflect environmental and management conditions, including the presence of endosymbionts (*Wolbachia* and *Spiroplasma*), and the treated versus untreated areas, although these factors were not directly quantified in this study. Given the agricultural impact of *D. opuntiae* infestations in Morocco, understanding its microbial diversity provides valuable insights into potential biological interactions that could be leveraged for control. These findings lay the groundwork for developing microbiome-informed pest management approaches, such as exploiting symbiont-based strategies or manipulating key microbial taxa to reduce pest fitness or transmission potential [162,163]. 

## 5. Conclusions

This study revealed the complex bacterial community structure of *D. opuntiae*, which is shaped by geography, sex, and developmental stage. Key findings include: (1) first detection of *Wolbachia* in *D. opuntiae* with geographic variation in prevalence; (2) male-specific *Spiroplasma* infection; (3) dominance of *Candidatus* Dactylopiibacterium across all populations; (4) hidden diversity revealed after removing the dominant symbiont; and (5) sex-specific microbiome specialization, with females harboring more restricted communities. The complementary use of targeted PCR and MinION sequencing proved to be essential for comprehensive characterization. These findings provide a foundation for developing microbiome-based pest management strategies, particularly through potential *Wolbachia* transinfection approaches or the disruption of obligate symbionts.

## Figures and Tables

**Figure 1 insects-16-01184-f001:**
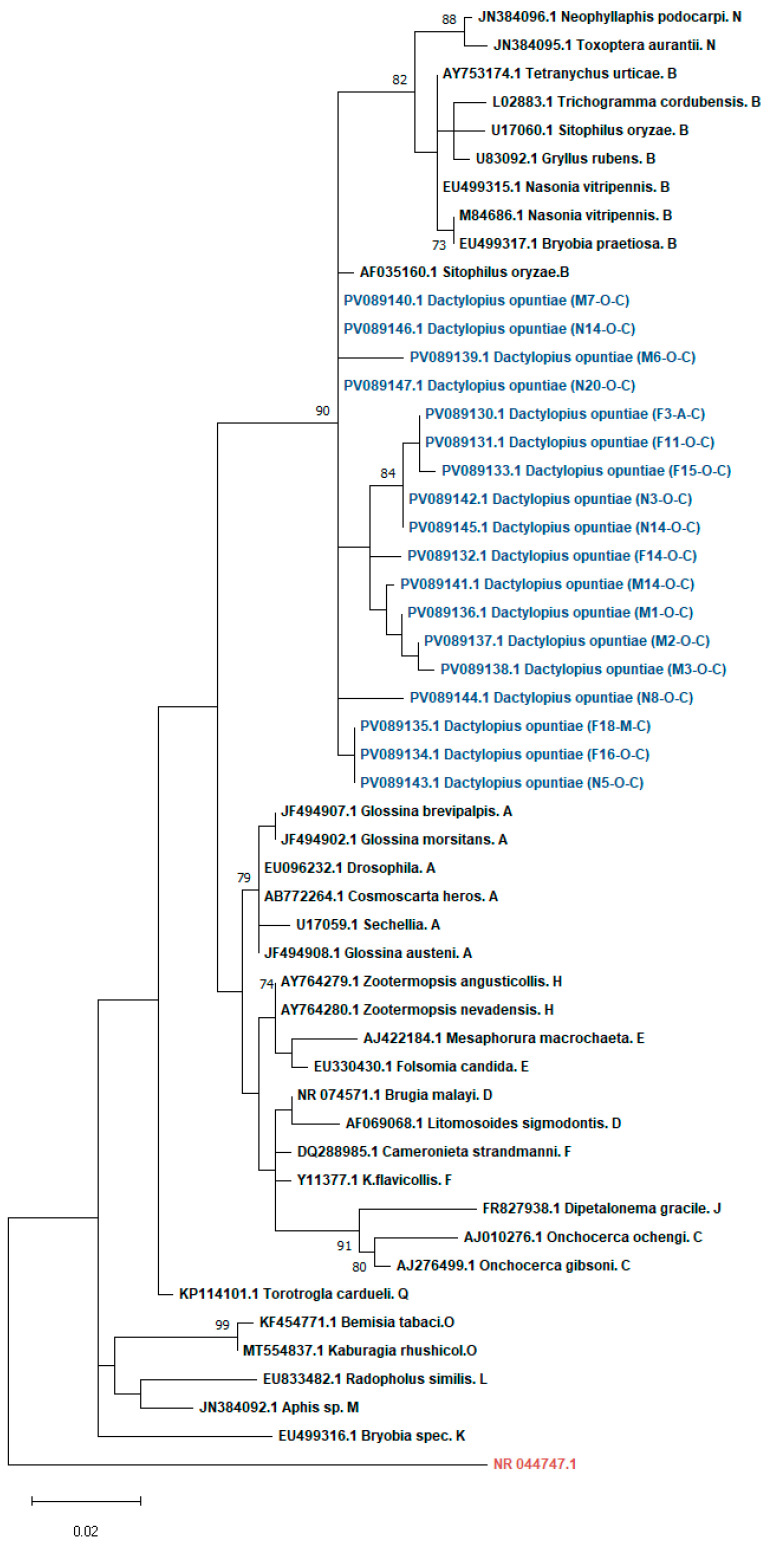
Maximum Likelihood phylogenetic tree of *Wolbachia* 16S rRNA sequences (353 bp) amplified from *Dactylopius opuntiae* samples. The sequences obtained in this study (GenBank accession numbers PV089130–PV089147) are highlighted in blue. Samples were coded by gender and developmental stage (M: male, F: female, N: nymph) and collection site (A: Agadir, O: Ouazzane, M: Meknes) (C: Cochineal). Reference sequences were retrieved from GenBank, including *Wolbachia* sequences representing multiple supergroups (B–H, K–O, Q) (black) and *Ehrlichia ewingii* as an outgroup (red). The tree was constructed using the Maximum Likelihood method with bootstrap values based on 1000 replicates (only values above 70% are shown).

**Figure 2 insects-16-01184-f002:**
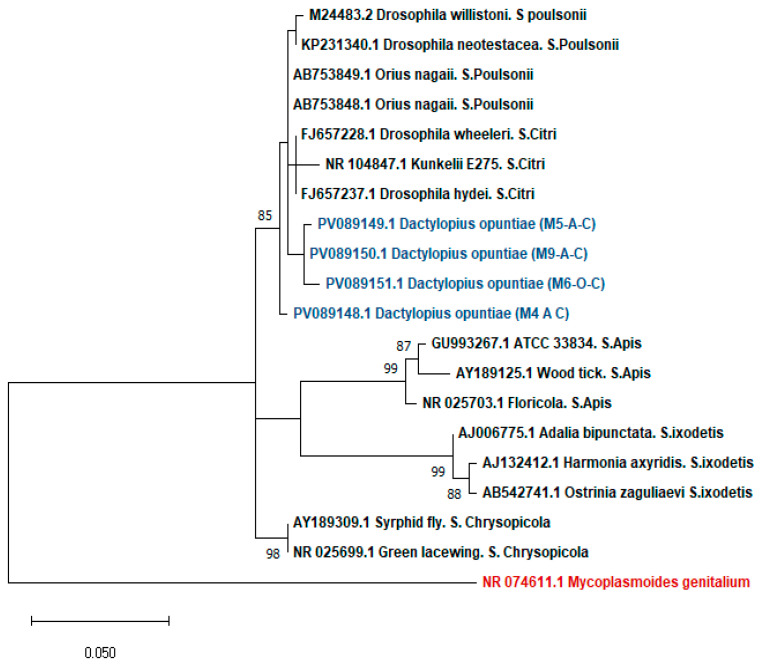
Maximum Likelihood phylogenetic tree of the four *Spiroplasma* 16S rRNA gene sequences (349 bp) amplified from *D. opuntiae*. The sequences obtained in this study (GenBank accession numbers PV089148–PV089151) are highlighted in blue. Sample codes: M (male), A (Agadir), O (Ouazzane), and C (Cochineal). Reference sequences representing the major *Spiroplasma* groups were included: Poulsonii, Citri, Chrysopicola, Ixodetis, and Apis groups (black). The outgroup sequences are shown in red. GenBank accession numbers and host species are indicated for each sequence in the table. Bootstrap support values (>70%) were based on 1000 replicates.

**Figure 3 insects-16-01184-f003:**
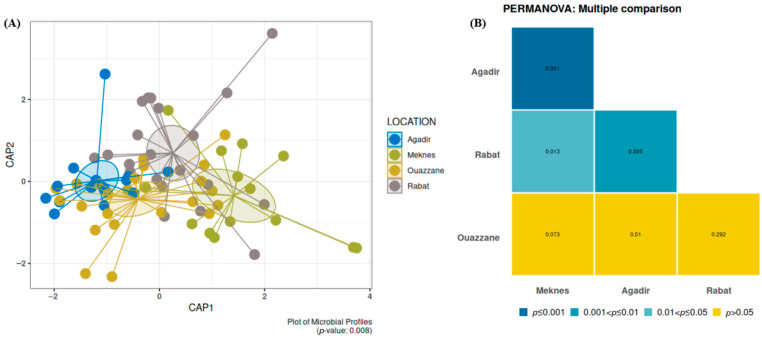
Diversity of *D. opuntiae*-associated bacterial communities based on the sites of the Constrained Analysis of Principal Coordinates (CAP) plot using the Bray–Curtis metric (*p* < 0.008) (**A**) and pairwise comparison between locations (**B**).

**Figure 4 insects-16-01184-f004:**
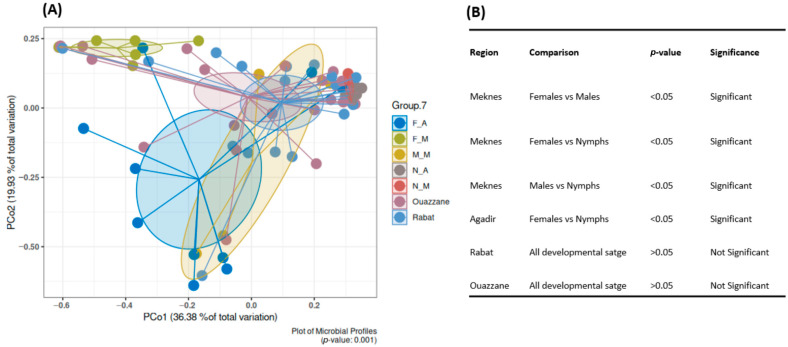
Beta diversity analysis based on Bray–Curtis metric of bacterial community associated with *D. opuntia* samples according to the regions and developmental stages, excluding *Candidatus*-Dactylopiibacterium. (**A**) Principal Coordinates Analysis (PCoA) plot. (**B**) Pairwise PERMANOVA results.

**Figure 5 insects-16-01184-f005:**
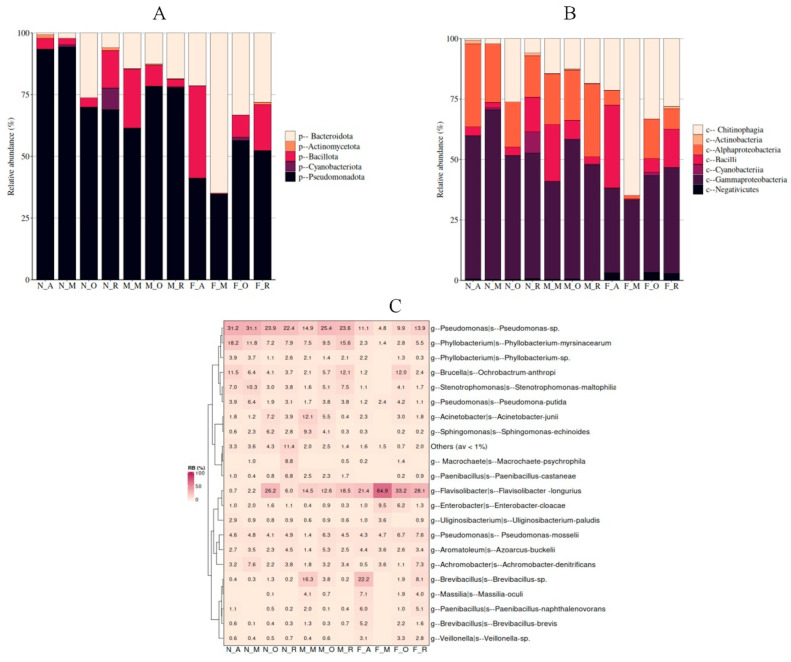
Relative abundance of bacteria associated with natural *Dactylopius opuntiae* population at the phylum (**A**), class level (**B**), and heat map (**C**) of bacterial genera and species identified in *D. opuntiae* populations after excluding *Candidatus* Dactylopiibacterium based on their developmental stages and locations.

**Table 1 insects-16-01184-t001:** Number of *Dactylopius opuntiae* adults and nymphs used for bacterial community analysis per location.

Region	Coordinates			Collection Date	Treatment	No. of Insects		
	Latitude	Longitude	Temperature			N	F	M
Agadir	30.206132	−9.534147	20 °C	Oct. 2022	Insecticide	5	10	-
Rabat	34.002736	−6.748109	22 °C	Nov. 2022	Burned	10	6	5
Meknes	33.963659	−5.576012	24 °C	Sept. 2022	Burned	4	7	4
Ouazzane	34.807449	−5.658892	18 °C	Dec. 2022	No	6	9	7

## Data Availability

Data to support this study are available from the National Center for Biotechnology Information (https://www.ncbi.nlm.nih.gov, accessed on 15 November 2025). The GenBank numbers are PV089130-PV089147 and PV089148-PV089151. The BioProject number is PRJNA1293817.

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
