# Peer review of "Analysis of the Bacterial Microbiota in Wild Populations of Prickly Pear Cochineal, Dactylopius opuntiae in Morocco"

_insects, 2025, doi:10.3390/insects16121184_

Round 1
Reviewer 1 Report
Comments and Suggestions for Authors
The prickly pear cochineal is a dangerous pest worldwide, incurring economical losses to edible cactus production industry. Just like other representatives of scale insects, it covers its body with waxy envelop which protect its from natural enemies and insecticide treatment. Detailed research of its biology is required to further facilitate development of control strategies. These studies should include microbiome research, as bacterial symbionts, both intestinal and intracellular, play crucial roles in physiology of insects by regulating nutrition, immunity, and reproduction. This makes the paper actual and suitable for the Insects journal. Modern approaches including high throughput sequencing and robust bioinformatics correspond to the up-to-date standards of sound research in this field. Sampling specimens from different regions allows tracking spatial variation of bacterial loads, including the influence of . Comparing between developmental stages and sexes further discloses patterns of symbiont distribution.
The paper is clearly written, the introduction sufficiently covers the topic and substantiates the goal, the methodology is given in sufficient detail. The results are straightforward, the discussion is relevant and the conclusions are convincing (but see further)
The work will be interesting to the journal’s audience and can be recommended for publication after correction of points listed below.
Line 17: I recommend using common species name in the simple summary alongside with Latin epithets
Line 21: the term “endosymbionts” is preferred over just “symbionts” to stress their intracellular localization
Line 22: it is not clear what is implied by “removal” – experimental eradication of the bacterium or processing of bioinformatics data
Line 33: Wolbachia which belongs to supergroup B is placed there not by analysis, the analysis only shows its place here
Line 43: “did” can be omitted
Lines 43-44: INSIGHTS????
Line 52: morphological description is of little relevance to the study
Line 57: “identical” to what?
Line 64: erratic
Lines 127-137: you meticulously describe treatment approaches and sampling, but how it relates to the study?
Lines 138-141: in spite of intention to “to evaluate whether the presence or absence of chemical treatments may affect the bacterial composition”, it is not mentioned further in the manuscript
Table 1. Neither burning, nor insecticide treatment are discussed in the manuscript
Line 154: is there an explanation why only two particular genera of inherited endosymbiotic bacteria were chosen for screening?
Lines 242-243: no need to repeat “reproductive symbionts” twice in a row. I would also argue that Spiroplasma is obligately a reproductive symbiont.
Line 243 and elsewhere: D. opuntiae flies – those are not flies
Lines 271-272: full-length full-16S – is the second “full” necessary?
Line 279: Uliginosibacterium represented by Candidatus Dactylopiibacterium – what does it mean?
Line 279: what’s the reason for underlining the bacterium name?
Lines 283-284: according to Table 1, these four regions are different in terms of control treatments. Could that be the reason for variation, and not geography itself?
Figures 1 & 2: for ML, there’s no need showing bootstrap support values below 70
Line 300: Wolbachia reference sequences from GenBank represent supergroup and outgroup sequences – what does it mean?
Line 308: there is only one outgroup
Lines 314-319: the sentence is lengthy, but can be easily broken down
Line 432: what is “population microbiota”?
Lines 437-443: no need repeating results in Discussion
Line 446: you indicate that Spiroplasma are found in males only but do not discuss possible reasons for that. Is it consistent with what is known about biology of this symbiont, or could that be just random variation due to low prevalence?
Lines 459-460: these lines repeat the text from Lines 446-447
Lines 473-474: what’s the reason mentioning these particular host taxa?
Line 484: is it erroneous use of italics?
Lines 493-494: the phrasal structure is obscure
Line 519: D. daci – what is implied by this taxon name? Is it the first mentioning of this genus in text
Line 540: interaction between ontogeny and geography – what is nature of this interaction?
Line 548: improper punctuation
Lines 551-552: host stage effect – why calling it stage effect if both nymphs and adult males display a consistent dominance? Which is the reason of host stage influencing the bacterial dominance?
Lines 559-560: associations with plants – please explain
Lines 564-565: check italics use
Lines 610-611: have you demonstrated effect of presence of Wolbachia and Spiroplasma on bacterial community?
Lines 611-614: influence of D. opuntiae on agriculture … highlights .. importance of … microbial diversity – this connection is not obvious
Lines 614-615: this is not obvious from the information provided in the manuscript what are the “microbiome-oriented integrated pest management strategies”
The use of commas as delineating marks should be checked throughout – it is used for decimals and thousands
Author Response
Reviewer 1
We are thankful to the reviewers for their helpful input and comments on our paper. We greatly appreciate the time and energy they spent reviewing our manuscript. We have taken into account the suggestions provided by the reviewers and have made the appropriate revisions to our paper in order to incorporate most of their feedback. To assist with the review process, we have marked all the alterations we made to the manuscript using track changes.
Comment 1: Line 17: I recommend using common species name in the simple summary alongside with Latin epithets.
Response: The common species name “cochineal” has been added alongside the Latin name in the Simple Summary section (Line 17).
Comment 2: Line 21: the term “endosymbionts” is preferred over just “symbionts” to stress their intracellular localization
Response: The term “symbionts” has been replaced with “endosymbionts” to emphasize their intracellular localization (Line 21).
Comment 3: Line 22: it is not clear what is implied by “removal” – experimental eradication of the bacterium or processing of bioinformatics data
Response: To clarify, “removal” refers to the exclusion of Candidatus Dactylopiibacterium sequences during bioinformatic processing. The text has been revised accordingly (Line 23).
Comment 4: Line 33: Wolbachia which belongs to supergroup B is placed there not by analysis, the analysis only shows its place here
Response: We thank the reviewer for this comment. The sentence has been revised to clarify that the phylogenetic analysis indicates that Wolbachia sequences likely belong to supergroup B, based on similarity to reference sequences (Line 34-35).
Comment 5: Line 43: “did” can be omitted
Response: The word “did” has been removed to improve sentence clarity and flow (Line 45).
Comment 6: Lines 43-44: INSIGHTS????
Response: We thank the reviewer for this comment. The sentence has been revised to clarify the type of insights provided, specifically emphasizing the roles of endosymbionts in host adaptation and their relevance to microbiome-based pest management (Line 47-48).
Comment 7: Line 52: morphological description is of little relevance to the study
Response: We thank the reviewer for this remark. However, we preferred to retain the brief morphological description to provide context for readers who may not be familiar with D. opuntiae, as this helps introduce the biological characteristics of the studied species.
Comment 8: Line 57: “identical” to what?
Response: We thank the reviewer for this observation. To clarify the context and maintain focus on the Moroccan samples, the mention of other regions (including Brazil) has been removed from lines (Line 61-63), as it was not directly relevant to the scope of this study.
Comment 9: Line 64: erratic
Response: The term “erratic” was replaced with “uncontrollable” to better describe the rapid and widespread nature of infestations (Line 68)
Comment 10: Lines 127-137: you meticulously describe treatment approaches and sampling, but how it relates to the study?
Response: We thank the reviewer for this observation. The description of management practices (burning, chemical treatment, and untreated areas) has been revised to be more concise in the Materials and Methods. These details are provided to give context about the sampling environments, while the analysis primarily focuses on geographical variation in the microbiome composition of D. opuntiae (Line 138-143).
Comment 11: Lines 138-141: in spite of intention to “to evaluate whether the presence or absence of chemical treatments may affect the bacterial composition”, it is not mentioned further in the manuscript
Response: We thank the reviewer for this observation. The Materials and Methods section (Line 138-143) has been revised to describe management practices (burning, chemical treatment, and untreated areas) in a concise manner without implying that treatment effects were the primary focus. In the Results section, we report that bacterial communities from the chemically treated region (Agadir) were similar to those from the untreated region (Ouazzane), indicating that chemical treatment had a limited influence on microbiome composition. Therefore, the analysis focuses on geographical variation rather than treatment effects (Line 324-330).
Comment 12: Table 1. Neither burning, nor insecticide treatment are discussed in the manuscript
Response: We thank the reviewer for pointing this out. Details of the burned (Rabat and Meknes) and chemically treated (Agadir) regions have been added to the Results and Discussion sections. Our analyses show that bacterial communities from burned sites differed from those in the chemically treated region, indicating that management practices may influence microbiome composition. However, differences between the two burned regions themselves highlight the predominance of geographical factors over treatment effects. These observations are now clearly described in both the Results and Discussion sections, with supporting references included (Line 324-330) and (Line 592-604).
Comment 13: Line 154: is there an explanation why only two particular genera of inherited endosymbiotic bacteria were chosen for screening?
Response: We thank the reviewer for this insightful comment. Wolbachia and Spiroplasma were specifically targeted based on prior investigations in Dactylopius species. While Spiroplasma has been previously reported in D. opuntiae, Wolbachia has not. Consequently, our screening was designed both to confirm the presence of Spiroplasma and to explore, for the first time, the potential occurrence of Wolbachia in D. opuntiae. To clarify the choice in the manuscript. we included the following paragraph (Line 172-177).
Comment 14: Lines 242-243: no need to repeat “reproductive symbionts” twice in a row. I would also argue that Spiroplasma is an obligately a reproductive symbiont.
Response: We thank the reviewer for this valuable observation. We corrected the redundancy and replaced “reproductive symbionts” with “endosymbionts” to accurately describe Spiroplasma and Wolbachia, which are not exclusively associated with reproductive functions. The manuscript has been revised accordingly (Line 266-269).
Comment 15: Line 243 and elsewhere: D. opuntiae flies – those are not flies
Response: We have corrected the text to replace “flies” with “samples” to accurately reflect that the tested material consisted of D. opuntiae scale insect rather than flies (Lines 269, 273, 278, 279, 360).
Comment 16: Lines 271-272: full-length full-16S – is the second “full” necessary?
Response: We thank the reviewer for noticing this typographical error. The sentence has been corrected to remove the redundant “full-.” (Line 298).
Comment 17: Line 279: Uliginosibacterium represented by Candidatus Dactylopiibacterium – what does it mean?
Response: We thank the reviewer for pointing this out. The sentence has been revised to clarify that Uliginosibacterium is the genus, and Candidatus Dactylopiibacterium is the species (Line 305).
Comment 18: Line 279: what’s the reason for underlining the bacterium name?
Response: Underlining has been removed, and the names are now properly italicized (Line 305).
Comment 19: Lines 283-284: according to Table 1, these four regions are different in terms of control treatments. Could that be the reason for variation, and not geography itself?
Response: We thank the reviewer for this insightful comment. As discussed in our responses to Comments 10–12, the study design considered both management practices (burning, chemical treatment, and untreated conditions) and geographical variation. However, the comparative analyses indicated that chemical treatments had a limited influence on the bacterial communities of D. opuntiae, whereas geographical factors were the main drivers of variation among regions. These aspects have been clarified in the revised Results and Discussion sections.
Comment 20: Figures 1 & 2: for ML, there’s no need showing bootstrap support values below 70
Response: We appreciate the reviewer’s suggestion. We have updated Figures 1 and 2 by removing all bootstrap support values below 70%, as recommended.
Comment 21: Line 300: Wolbachia reference sequences from GenBank represent supergroup and outgroup sequences – what does it mean?
Response: We thank the reviewer for this comment. The sentence was unclear. We have revised the caption of Figure 1 as follows: “Reference sequences were retrieved from GenBank, including Wolbachia sequences representing multiple supergroups (B–H, K–O, Q) (black) and Ehrlichia ewingii as an outgroup (red) (Line 336-339).
Comment 22: Line 308: there is only one outgroup
Response: Using one closely related member of Anaplasmataceae family (Ehrlichia ewingii) minimizes the alignment artifact and long branch attraction, which might arise if multiple or distantly related outgroups are used.
Comment 23: Lines 314-319: the sentence is lengthy, but can be easily broken down
Response: The sentence has been revised to improve clarity and readability by breaking it into shorter, more concise statements while maintaining the accuracy of the reported values (Line 352-356).
Comment 24: Line 432: What is “population microbiota”?
Response: We thank the reviewer for this observation. The figure caption has been corrected (Line 471).
Comment 25: Lines 437-443: no need repeating results in Discussion
Response: We thank the reviewer for this valuable observation. The introductory part of the discussion has been revised to remove redundancy with the results section. The updated version now focuses on the interpretation and broader biological significance of the findings rather than restating methodological or descriptive details (Line 476-481).
Comment 26: Line 446: you indicate that Spiroplasma are found in males only but do not discuss possible reasons for that. Is it consistent with what is known about biology of this symbiont, or could that be just random variation due to low prevalence?
Response: We thank the reviewer for this insightful comment. The observation that Spiroplasma was detected only in a subset of males may reflect either low prevalence or a possible sex-specific association. As our study did not include targeted analyses to test this hypothesis, further investigation would be necessary to determine whether this pattern represents a true biological phenomenon or is due to limited sample detection (Line 491-495).
Comment 27: Lines 459-460: these lines repeat the text from Lines 446-447
Response: We thank the reviewer for highlighting this repetition. We have revised the text to remove redundancy (Line 507-509).
Comment 28: Lines 473-474: what’s the reason mentioning these particular host taxa?
Response: We thank the reviewer for this observation. The mention of aphids, Drosophilidae, and other arthropod taxa is intended to provide context for the prevalence and ecological significance of Wolbachia infections across diverse insect hosts, highlighting that Wolbachia is widespread among arthropods and not unique to Dactylopius species.
Comment 29: Line 484: is it erroneous use of italics?
Response: We thank the reviewer for pointing this out. The phrase has been corrected in the revised manuscript (Line 533).
Comment 30: Lines 493-494: the phrasal structure is obscure
Response: We thank the reviewer for this comment. The sentence has been restructured to improve clarity and readability, separating previous metagenomic studies from the recent genomic analyses (Line 542-546).
Comment 31: Line 519: D. daci – what is implied by this taxon name? Is it the first mentioning of this genus in text
Response: We thank the reviewer for this comment. The full species name, Dipterophagus daci, has now been provided upon first mention in the text, with proper italicization according to taxonomic conventions (Line 575).
Comment 32: Line 540: interaction between ontogeny and geography – what is nature of this interaction?
Response: We thank the reviewer for this comment. In this section (Line 607-611), we discussed mainly the variation observed between genders and developmental stages. Therefore, we have modified the text to clarify this point.
Comment 33: Line 548: improper punctuation
Response: Thank you for noting this. The punctuation has been corrected by replacing the period with a comma to ensure grammatical accuracy. (Line 618)
Comment 34: Lines 551-552: host stage effect – why calling it stage effect if both nymphs and adult males display a consistent dominance? Which is the reason of host stage influencing the bacterial dominance?
Response: We revised the text to clarify that although both nymphs and males showed similar Pseudomonas dominance across the regions, this contrasts with females, whose bacterial profiles varied geographically (Line 618-623).
Comment 35: Lines 559-560: associations with plants – please explain
Response: We thank the reviewer for the comment. The statement refers to the fact that low-abundance genera such as Phyllobacterium and Stenotrophomonas are commonly found in soil and plant-associated environments. Their detection in D. opuntiae likely reflects incidental acquisition from the host plant or the surrounding environment rather than stable, insect-specific associations (Line 630-633).
Comment 36: Lines 564-565: check italics use
Response: The font style has been corrected in the revised manuscript (Line 637-638).
Comment 37: Lines 610-611: have you demonstrated effect of presence of Wolbachia and Spiroplasma on bacterial community?
Response: We thank the reviewer for this comment. While our study identified the presence of Wolbachia and Spiroplasma and characterized overall bacterial community composition, we did not directly test or quantify the effects of these reproductive symbionts on the broader microbiota. The sentence has been revised to clarify that (Line 683-686).
Comment 38: Lines 611-614: influence of D. opuntiae on agriculture … highlights .. importance of … microbial diversity – this connection is not obvious
Response: We thank the reviewer for this valuable comment. The sentence has been revised to clarify the logical connection between D. opuntiae’s agricultural impact and the relevance of microbial diversity. The revised text now emphasizes that understanding the microbiota of D. opuntiae can inform the development of microbiome-based pest management strategies, such as exploiting or manipulating symbiotic bacteria to reduce pest fitness. making the connection between agricultural impact and microbial diversity explicit (Line 690-696).
Comment 39: Lines 614-615: this is not obvious from the information provided in the manuscript what are the “microbiome-oriented integrated pest management strategies”
Response: We thank the reviewer for this comment. This point has been clarified in the revised text (Line 690-696), as described in our response to Comment 38. The updated section now explicitly explains that microbiome-oriented strategies refer to potential pest management approaches based on manipulating or exploiting symbiotic bacteria to reduce pest fitness.
Comment 40: The use of commas as delineating marks should be checked throughout – it is used for decimals and thousands
Response: We appreciate the reviewer’s observation. All numerical values have been reviewed, and commas used as decimal separators were replaced to ensure consistency with standard scientific formatting.

Reviewer 2 Report
Comments and Suggestions for Authors
This paper analyzes the bacterial microbiome of Dactylopius opuntiae. There has been some data on this (Martinez-Martinez 2021 and a preprint 2024) but not much, and not from Morocco where this species is invasive.
I noticed the above two papers were not in the references. You will need to cite them!
This one is published:
Martínez‐Martínez, Susana, Esteban Rodríguez‐Leyva, Sergio Aranda‐Ocampo, Ma Teresa Santillán‐Galicia, Antonio Hernández‐López, and Ariel W. Guzmán‐Franco. "Bacteria associated with carminic acid metabolism in the intestinal tract of three predators of Dactylopius opuntiae." Entomologia Experimentalis et Applicata 172, no. 2 (2024): 183-192.
^It used cultivable micorbes, not 16S, so your work will uncover more microbes.
This one is a pre-print, but nonetheless you may need to look at it: https://doi.org/10.21203/rs.3.rs-384884/v1
I applaud the authors for sampling from multiple locations. Excellent.
The authors state in the abstract that "Core microbiome analysis revealed no universal genera across all group," but wouldn't Dactylopiibacterium be this core microbe? A big question I have throughout this paper is: how important are the 2.3% of non Dactylopiibacterium to the biology of the insect? Some of them may indeed be important, sure, but we don't know.
Are the other microbes gut microbes or surface microbes or something else?
117 replace "aimed to elucidate" with "investigated"
For the methods, you did not describe how you did core microbiome analysis other than to say you used MetaXplore. Are there more details, or is that a typical statistical test / program that MetaXplore runs?
Throughout the paper: I suspect this journal uses . instead of , for decimals, so change things like 16,6% into 16.6%
245 You can delete "Only D. opuntiae samples from Ouazzane, Meknes, and Agadir were infected with Wolbachia." since the next sentence says the same thing but better.
249 & 252 "In contrast" is not appropriate here, as the results do not seem to contrast the preceding results. Delete.
you write "The bacterial community of D. opuntiae varied across different regions" but also "After excluding the dominant Candidatus Dactylopiibacterium, no significant difference was observed between the regions." So did it vary or not? Were insects from one region more likely to have non-Dactylopiibacterium than others?
547 why this extra blank line?
Author Response
Reviewer 2:
We are grateful to the reviewer for his valuable comments and constructive feedback on our manuscript. We sincerely appreciate the time and effort he dedicated to carefully reviewing our work. We have thoroughly considered all his suggestions and made the necessary revisions to address his concerns. All changes made in the manuscript have been highlighted using track changes to facilitate the review process.
Comment 1: This paper analyzes the bacterial microbiome of Dactylopius opuntiae. There has been some data on this (Martinez-Martinez 2021 and a preprint 2024) but not much, and not from Morocco where this species is invasive. I noticed the above two papers were not in the references. You will need to cite them!
This one is published: Martínez‐Martínez, Susana, Esteban Rodríguez‐Leyva, Sergio Aranda‐Ocampo, Ma Teresa Santillán‐Galicia, Antonio Hernández‐López, and Ariel W. Guzmán‐Franco. "Bacteria associated with carminic acid metabolism in the intestinal tract of three predators of Dactylopius opuntiae." Entomologia Experimentalis et Applicata 172, no. 2 (2024): 183-192.
It used cultivable microbes, not 16S, so your work will uncover more microbes.
This one is a pre-print, but nonetheless, you may need to look at it: https://doi.org/10.21203/rs.3.rs-384884/v1
Response: We thank the reviewer for bringing these important references to our attention. Both Martínez-Martínez et al. (2023) and the preprint by Rodríguez-Leyva et al. (2021) have now been cited in the Introduction of the revised manuscript (Lines 101-107)
Comment 2: I applaud the authors for sampling from multiple locations. Excellent. The authors state in the abstract that "Core microbiome analysis revealed no universal genera across all group," but wouldn't Dactylopiibacterium be this core microbe?
Response: We thank the reviewer for this observation. While Candidatus Dactylopiibacterium was indeed the dominant and most prevalent taxon across all samples. We did not include it as part of the “core microbiome” in our analysis. Core microbiome definitions in microbial ecology typically focus on taxa that are consistently present and potentially functionally relevant across samples, while excluding obligate or highly dominant symbionts that could bias the analysis (Neu et al., 2021; Risely, 2020) [1,2]. Accordingly, the statement in the abstract refers to non-obligate bacterial genera, for which no universal taxa were detected across all groups.
Comment 3: A big question I have throughout this paper is: how important are the 2.3% of non Dactylopiibacterium to the biology of the insect? Some of them may indeed be important, sure, but we don't know.
Response: We thank the reviewer for this comment. While Candidatus Dactylopiibacterium dominates the microbiome of D. opuntiae, the remaining 2.3% of taxa, although low in abundance, may still contribute to host biology through interactions with the host or other microbes. However, their specific functional roles remain uncharacterized in this study. We have now added a statement in the Discussion acknowledging that these rare taxa may be ecologically or physiologically relevant, and that future functional analyses are needed to determine their significance (Line 563-567).
Comment 4: Are the other microbes gut microbes or surface microbes, or something else?
Response: We thank the reviewer for this comment. Before DNA extraction, all D. opuntiae individuals were surface-sterilized using 70% ethanol and rinsed with sterile water. DNA was then extracted from the entire insect, so the resulting microbiome profiles reflect both gut- and tissue-associated bacteria, while minimizing contributions from surface microbes. We have now added a clarification about that in the materials and methods section (Line 158-160).
Comment 5:117 replace "aimed to elucidate" with "investigated"
Response: We thank the reviewer for this suggestion. The text has been revised accordingly (Line 128).
Comment 6: For the methods, you did not describe how you did core microbiome analysis other than to say you used MetaXplore. Are there more details, or is that a typical statistical test / program that MetaXplore runs?
Response: We thank the reviewer for this comment. Core microbiome analyses were performed using MetaXplore (Bel Mokhtar et al., 2024)[3] to identify bacterial taxa consistently present across samples. Operational taxonomic units (OTUs) were considered part of the core if they were detected in ≥50% of the samples within each region (prevalence threshold) and had a relative abundance ≥0.1% (detection threshold). These criteria allowed us to focus on taxa that are consistently present across samples while excluding very rare OTUs (Line 241-246).
Comment 7: Throughout the paper, I suspect this journal uses. instead of, for decimals, so change things like 16,6% into 16.6%
Response: We thank the reviewer for this comment. All decimal points have been corrected throughout the manuscript.
Comment 8: 245 You can delete "Only D. opuntiae samples from Ouazzane, Meknes, and Agadir were infected with Wolbachia." since the next sentence says the same thing but better.
Response: We thank the reviewer for this suggestion. The sentence has been deleted from the text.
Comment 9: 249 & 252 "In contrast" is not appropriate here, as the results do not seem to contrast the preceding results. Delete.
Response: We thank the reviewer for this suggestion. ‘In contrast’ has been removed (Line 276).
Comment 10: you write "The bacterial community of D. opuntiae varied across different regions" but also "After excluding the dominant Candidatus Dactylopiibacterium, no significant difference was observed between the regions." So did it vary or not? Were insects from one region more likely to have non-Dactylopiibacterium than others?
Response: We thank the reviewer for this observation. The apparent variation in the bacterial community across regions is largely driven by the dominant symbiont, Candidatus Dactylopiibacterium. When this taxon is excluded from the analysis, no significant differences remain among regions, indicating that the regional variation is primarily attributable to differences in the relative abundance of this dominant species. Therefore, while the overall community composition varies across regions, the diversity of the low-abundance taxa is relatively consistent (Line 319-322).
Comment 11: 547 why this extra blank line?
Response: We thank the reviewer for pointing this out. The extra blank line has been removed in the revised manuscript.
References :
- Neu, A.T.; Allen, E.E.; Roy, K. Defining and Quantifying the Core Microbiome: Challenges and Prospects. Proceedings of the National Academy of Sciences 2021, 118, e2104429118, doi:10.1073/pnas.2104429118.
- Risely, A. Applying the Core Microbiome to Understand Host–Microbe Systems. Journal of Animal Ecology 2020, 89, 1549–1558, doi:10.1111/1365-2656.13229.
- Bel Mokhtar, N.; Asimakis, E.; Galiatsatos, I.; Maurady, A.; Stathopoulou, P.; Tsiamis, G. Development of MetaXplore: An Interactive Tool for Targeted Metagenomic Analysis. Current Issues in Molecular Biology 2024, 46, 4803–4814, doi:10.3390/cimb46050289.
